# Sustainability of AI-Assisted Mental Health Intervention: A Review of the Literature from 2020–2025

**DOI:** 10.3390/ijerph22091382

**Published:** 2025-09-04

**Authors:** Danicsa Karina Espino Carrasco, María del Rosario Palomino Alcántara, Carmen Graciela Arbulú Pérez Vargas, Briseidy Massiel Santa Cruz Espino, Luis Jhonny Dávila Valdera, Cindy Vargas Cabrera, Madeleine Espino Carrasco, Anny Dávila Valdera, Luz Mirella Agurto Córdova

**Affiliations:** 1School of Nursing, Faculty of Health Sciences, Universidad César Vallejo, Chiclayo 14000, Peru; carbulu@ucvvirtual.edu.pe (C.G.A.P.V.); eespinoca@ucvvirtual.edu.pe (M.E.C.); lagurtoco@ucvvirtual.edu.pe (L.M.A.C.); 2School of Nursing, Faculty of Health Sciences, Universidad Particular de Chiclayo, Chiclayo 14000, Peru; mapal.udch@outlook.es; 3School of Nursing, Faculty of Health Sciences, Universidad Señor de Sipán, Chiclayo 14000, Peru; sespinobriseidm@uss.edu.pe (B.M.S.C.E.); vcabrera@uss.edu.pe (C.V.C.); 4School of Nursing, Faculty of Health Sciences, Universidad Nacional Mayor de San Marcos, Lima 00051, Peru; ddavilaval@ucvvirtual.edu.pe (L.J.D.V.); ankat20@hotmail.com (A.D.V.)

**Keywords:** artificial intelligence, sustainable mental health, ethics, resource efficiency, personalization, cultural adaptation, human-AI integration

## Abstract

This systematic review examines the role of artificial intelligence (AI) in the development of sustainable mental health interventions through a comprehensive analysis of literature published between 2020 and 2025. In accordance with the PRISMA guidelines, 62 studies were selected from 1652 initially identified records across four major databases. The results revealed four dimensions critical for sustainability: ethical considerations (privacy, informed consent, bias, and human oversight), personalization approaches (federated learning and AI-enhanced therapeutic interventions), risk mitigation strategies (data security, algorithmic bias, and clinical efficacy), and implementation challenges (technical infrastructure, cultural adaptation, and resource allocation). The findings demonstrate that long-term sustainability depends on ethics-driven approaches, resource-efficient techniques such as federated learning, culturally adaptive systems, and appropriate human-AI integration. The study concludes that sustainable mental health AI requires addressing both technical efficacy and ethical integrity while ensuring equitable access across diverse contexts. Future research should focus on longitudinal studies examining the long-term effectiveness and cultural adaptability of AI interventions in resource-limited settings.

## 1. Introduction

The integration of artificial intelligence (AI) in mental health care presents significant sustainability challenges within the international context despite its promising applications. Sustainability concerns arise from multiple interconnected factors that threaten the long-term viability and scalability of AI mental health interventions. Mental health professionals encounter various barriers when adopting AI-driven interventions, including difficulties integrating these tools into established clinical workflows, which undermines the economic sustainability of implementation by requiring extensive retraining and system modifications [1,2]. A qualitative study by [3] revealed that mental health practitioners express concerns about the impact of AI technologies on the therapeutic relationship, specifically regarding the potential loss of human connection essential for effective treatment, raising questions about the social sustainability of AI-enhanced care models.

These concerns are further compounded by ethical dilemmas surrounding data privacy and security, particularly in low- and middle-income countries where comprehensive regulatory frameworks are lacking, creating environmental and social sustainability risks through potential exploitation of vulnerable populations [4,5]. The economic unsustainability of current approaches is evident in research by [6], who highlighted the limited empirical evidence supporting the efficacy of AI-driven mental health applications, noting that many interventions lack rigorous evaluation through randomized controlled trials. This creates a precarious situation where substantial financial investments in AI technologies are deployed without sufficient understanding of their long-term effectiveness or potential unintended consequences, threatening both economic viability and clinical sustainability.

Additionally, the environmental sustainability of AI mental health interventions faces challenges from high computational resource requirements, energy consumption for model training and deployment, and the digital divide that excludes populations without adequate technological infrastructure. Social sustainability is compromised by algorithmic bias that perpetuates existing healthcare disparities, a lack of cultural adaptation in AI systems, and the potential displacement of human mental health workers without adequate transition planning. These sustainability challenges necessitate systematic examination to understand how AI can be responsibly integrated into mental health care while ensuring long-term viability across economic, environmental, and social dimensions.

A critical analysis of the current literature reveals significant knowledge gaps in understanding how to achieve sustainable AI mental health interventions. Despite the proliferation of AI-based mental health tools, there remains a substantial disconnect between technological development and sustainable clinical implementation. Interdisciplinary collaboration between AI developers and mental health professionals is notably insufficient, leading to technologies that may not adequately address the practical needs of clinicians or patients while failing to consider long-term sustainability requirements [1,3]. This gap is further evidenced by research from [7], who reported that user engagement and perceptions remain underexplored factors in the design and implementation of AI mental health interventions, critical components for ensuring social and economic sustainability.

Another crucial knowledge gap concerns the sustainability implications of cultural sensitivity and diversity in AI systems. As emphasized by [4,8], current AI models often lack diversity in training datasets, potentially leading to biased outcomes that fail to serve diverse populations effectively, undermining the social sustainability and scalability of these interventions. Previous studies by [9,10] have established the importance of contextually appropriate mental health interventions for achieving sustainability, yet research on culturally adaptive AI systems that can maintain effectiveness across diverse contexts remains scarce. The absence of comprehensive sustainability frameworks for evaluating AI mental health interventions creates additional gaps in understanding how to assess and ensure long-term viability, highlighting the urgency and relevance of addressing these knowledge gaps through a systematic review.

Understanding the intersection of AI and sustainable mental health interventions requires conceptual clarity regarding both constructs. Artificial intelligence represents a multidisciplinary field within computer science focused on creating systems capable of performing tasks that typically require human intelligence, including learning, reasoning, decision-making, and problem-solving [11,12,13,14]. AI systems can be broadly categorized into rule-based expert systems and machine learning systems [15], with the primary goal of enabling machines to exhibit intelligent behavior through advanced algorithms and computational models [12,16].

Sustainable mental health interventions, on the other hand, represent interventions that aim to improve mental health outcomes while ensuring long-term viability, economic feasibility, and integration into existing systems across diverse contexts. These interventions are critical for achieving the Sustainable Development Goals related to health and well-being [10,17,18]. Sustainability encompasses multiple dimensions: economic viability through cost-effectiveness and resource efficiency, environmental responsibility through minimized ecological footprint, and social equity through cultural appropriateness and community acceptance. Key strategies include task sharing, scaling up successful programs, addressing structural barriers, building organizational capacity, promoting mental well-being, and integrating mental health into broader development initiatives [9,10,19]. The integration of AI with these sustainability principles creates unique opportunities for enhancing intervention reach and effectiveness while addressing traditional barriers such as workforce shortages and resource constraints, forming the conceptual foundation that justifies our research questions and search methodology.

This review aims to conduct a scientific analysis of the role of artificial intelligence in sustainable mental health interventions by examining the literature from 2020–2025. The general objective is to perform a comprehensive scientific review analyzing the role of artificial intelligence in sustainable mental health interventions. Specifically, this study seeks to address the following research questions: (1) What are the ethical considerations in the use of AI for mental health interventions? (2) How can AI be used to personalize mental health interventions while respecting patient privacy? (3) What are the potential risks of using AI to personalize mental health interventions, and how can they be mitigated? (4) What are the challenges in implementing AI-based mental health interventions on a global scale? Additionally, (5) a bibliometric analysis of keywords and authors in the field is conducted. Investigating this intersection is necessary, as existing research tends to focus either on the technological aspects of AI or on traditional approaches to mental health care, with limited attention given to their sustainable integration. The novelty of this research lies in its comprehensive examination of how AI can contribute to the sustainability of mental health interventions while addressing critical ethical and implementation challenges identified in the literature. By synthesizing findings across disciplinary boundaries, this study provides insights into the development of more effective, equitable, and sustainable AI-driven mental health interventions, thereby addressing the knowledge gaps related to interdisciplinary collaboration, cultural sensitivity, and evidence-based implementation.

The theoretical and practical significance of this research is multifaceted. From a theoretical perspective, this review contributes to knowledge by synthesizing fragmented literature across the fields of artificial intelligence, mental health, and sustainability science. As refs. [2,6] indicate, there is a need for more comprehensive frameworks that integrate these disciplines effectively. This study helps refine conceptual models of how AI can sustainably support mental health interventions while respecting ethical boundaries and cultural contexts. From a practical standpoint, the findings will directly benefit mental health professionals seeking to incorporate AI tools into their practice by providing evidence-based guidelines for implementation. Organizations and policymakers will gain insights into addressing structural barriers to sustainable AI integration, which [10] identifies as crucial for intervention success. Additionally, the review’s emphasis on cultural sensitivity will help developers create more inclusive AI systems that can effectively serve diverse populations, addressing the concerns raised by [8] regarding cultural biases in existing technologies. By bridging theoretical understanding with practical applications, this research will facilitate more effective knowledge translation between AI development and clinical mental health practice.

## 2. Materials and Methods

This study employed a comprehensive two-phase approach to explore the role of artificial intelligence in sustainable mental health interventions. Initially, a bibliometric analysis was conducted to examine the evolution of research interest, identify prominent authors, and highlight the main trends in the field. An in-depth analysis of the theoretical contributions was subsequently performed, with a direct focus on the central theme of the systematic review.

Bibliometrics, which employs statistical techniques, allows for a quantitative evaluation of published scientific output, highlighting predominant research trends through indicators such as citations, authorship, and the impact of contributions. However, while bibliometric information is valuable, it was complemented with hermeneutic analysis to provide a deeper and more contextualized understanding of the topic under study.

The bibliometric approach was used as an initial exploration of scientific production, employing four major databases: Scopus, Web of Science (WOS), Science Direct, and Dimensions. Database searches were conducted between 15–20 January 2025, covering publications from January 2020 to December 2025. Scopus was selected as the primary database for the bibliometric analysis because of its multidisciplinary nature and the inclusion of various relevant publishers. To identify the critical mass of documents, specific search protocols were applied to each database using Boolean operators and controlled vocabulary terms, as detailed in Table 1. The search strategy employed a combination of title, abstract, and keyword fields to ensure comprehensive coverage while maintaining precision in document retrieval.

This systematic review focused on thoroughly analyzing the scientific evolution of AI applications in sustainable mental health interventions from 2020–2025. The inclusion criteria targeted studies that focused primarily on this topic, excluding works that addressed it tangentially or were not pertinent. To ensure the quality of the selected studies, aspects such as the clarity of research objectives in direct relation to their thematic focus, the robustness of methodological design, the presentation of relevant data and cases, the pertinence in addressing research questions, and the depth of result discussions were evaluated.

The exclusion criteria for this systematic literature review (SLR) were based on two fundamental requirements: thematic evaluation and methodological evaluation. The thematic evaluation considered whether the studies focused on objectives aimed at analyzing AI applications in sustainable mental health interventions, as well as whether they established a clear link between these technologies and improvements in mental health outcomes. The methodological evaluation was based on rigorous criteria, including whether the study design was oriented toward achieving these objectives, whether it presented empirical data supporting its results, whether the research questions were adequately addressed, and whether the results included a robust and coherent discussion. Studies that failed to meet these requirements were excluded, ensuring the quality and relevance of the selected documents.

To ensure methodological rigor, a systematic quality assessment of the selected studies was implemented via a standardized assessment tool based on the Critical Appraisal Skills Program (CASP). Each study was independently assessed by two investigators via a 12-point scale examining four main dimensions: (1) quality of the research design (clear objectives, appropriate methodology, robust design); (2) data collection and analysis (sampling methods, instrument validity, analysis procedures); (3) results and discussion (clear presentation, appropriate interpretation, discussion of limitations); and (4) relevance and contribution (contribution to knowledge, practical implications, future recommendations). Studies were classified as high quality (10–12 points), medium quality (7–9 points), or low quality (≤6 points), excluding those that scored below 7. Inter-rater reliability was assessed using Cohen’s kappa coefficient, yielding substantial agreement (κ = 0.78, 95% CI: 0.71–0.85). Discrepancies in the evaluation were resolved by discussion, and when necessary, a third investigator was consulted to reach a consensus. To address potential bias, we acknowledge limitations regarding publication bias favoring positive results and language bias due to our focus on English-language publications, which may have excluded relevant studies from non-English speaking regions where AI mental health research is emerging. Additional sources of bias were considered, including funding bias, as industry-sponsored studies may overrepresent commercially viable AI solutions while underreporting negative outcomes or implementation challenges. Geographic bias was also evident, with research predominantly originating from high-income countries, potentially limiting the generalizability of findings to resource-constrained settings where sustainable mental health interventions are most critically needed.

This research employed a structured approach to conduct a systematic literature review (SLR). It was divided into three main stages: (i) Systematic review strategy, which included defining the research questions, creating a search strategy, and identifying relevant publications; (ii) Conducting the search, during which systematic filters were applied according to inclusion and exclusion criteria, selecting only primary and analytical publications pertinent to the research topic; and (iii) results and discussions, where the analysis criteria were defined, a characterization framework was structured, and an exhaustive analysis of the findings was conducted. This process enabled the establishment of a rigorous methodological framework that ensured the precise selection and evaluation of AI applications in sustainable mental health interventions, as depicted in Figure 1.

The strategy implemented was based on the protocols of the Preferred Reporting Items for Systematic Reviews and Meta-Analyses Protocols (PRISMA), which are designed to outline well-founded methods with defined hypotheses and a structured plan for the Systematic Literature Review (SLR). This approach provides predefined methodological and analytical justifications, allowing for the avoidance of biases and arbitrary decisions throughout the review process. The PRISMA diagram for this study is presented in Figure 2.

From the initial 1652 records identified through database searching, 297 duplicates were removed, leaving 1355 records for screening. After applying the inclusion and exclusion criteria to titles and abstracts, 1108 records were excluded. The remaining 210 full-text articles were assessed for eligibility, with 95 articles excluded for reasons such as insufficient focus on AI applications in mental health, lack of sustainability focus, methodological limitations, or publication date outside the target range. This resulted in 62 studies for the final qualitative synthesis.

For bibliometric analysis, VOSviewer (version 1.6.18) was used to visualize the scientific landscape of the field. The cooccurrences of keywords, coauthorship networks, and citation patterns were analyzed to identify thematic clusters and influential research. The following parameters were applied in VOSviewer: for keyword analysis, a minimum occurrence of 5 was needed; for coauthorship network analysis, authors with at least 2 publications were included; and for citation analysis, a minimum of 10 citations per document was set as the threshold. The resulting visualizations were exported as network maps and density visualizations, providing insights into the conceptual structure and evolution of the field.

For the qualitative analysis of content, all 62 selected studies were imported into Microsoft Excel for systematic coding and categorization. A coding framework was developed on the basis of initial reading of the literature, which included the following dimensions: (1) type of AI technology employed; (2) mental health condition addressed; (3) intervention characteristics; (4) sustainability factors; (5) ethical considerations; (6) implementation challenges; and (7) reported outcomes. Each article was coded independently by two researchers, with a third researcher resolving any disagreements. Microsoft Word was used for thematic synthesis, where similar concepts and findings were grouped to identify overarching themes and subthemes across the literature.

To ensure transparency and traceability throughout the entire bibliometric and PRISMA research process, a link was provided granting access to all the data collected and analyzed. This resource offers a comprehensive view of the methodological process, facilitating verification, replication, and further analysis by other interested researchers.

The methodological approach employed in this study combines quantitative bibliometric analysis with qualitative content analysis, providing a comprehensive understanding of both the structural patterns of research production and the substantive content of scientific contributions in the field of AI for sustainable mental health interventions.

## 3. Results

### 3.1. Systematic Literature Review

Table 2 presents a systematic categorization of the 62 selected scientific articles, aligned with the general objective of analyzing the role of artificial intelligence in sustainable mental health interventions. A stratified distribution emerges across four key dimensions: ethical considerations (30 articles), personalization approaches (22 articles), risk and mitigation strategies (22 articles), and implementation challenges (16 articles). The predominance of studies focused on ethical aspects suggests a scientific acknowledgment of the need to establish solid normative frameworks prior to technological implementation. The subcategories identified within each primary dimension reveal a coherent taxonomy that enables a systematic approach to the study’s specific research questions, providing a structured foundation for the subsequent analysis of AI-based interventions in mental health contexts.

Table 3 provides a qualitative synthesis of the scientific contributions by category, directly addressing the four specific research questions. Regarding the first question on ethical considerations, critical findings emerge on privacy, informed consent, algorithmic bias, and human oversight. In relation to the second question on personalization and privacy, significant advancements are noted in federated learning and AI-powered therapeutic interventions. The third question, concerning risks and mitigation strategies, is addressed through analyses of data security vulnerabilities, algorithmic bias, clinical efficacy, and human-AI integration. Finally, for the fourth question on global implementation challenges, critical barriers are identified in terms of technical infrastructure, cultural adaptation, resource allocation, and evidence requirements. This structure allows for a comprehensive understanding of scientific findings within each analytical dimension, facilitating the identification of convergent and divergent patterns in the literature.

Table 4 presents a multidimensional analytical framework examining engagement aspects and outcomes across the included studies, complementing the general objective of the research. The analysis reveals distinctive patterns of engagement (technical, clinical, ethical, and user experience) and outcomes (treatment efficacy, implementation success, and sustainability) that vary systematically across categories. Ethical considerations predominantly demonstrate ethical engagement, with outcomes centered on trust and equity. Personalization approaches show balanced engagement across multiple dimensions, with a focus on efficacy and user experience outcomes. Risk mitigation strategies highlight a robust technical-clinical approach aimed at ensuring security and quality outcomes. Implementation challenges present complex engagement patterns, with particular attention to sustainability as a critical outcome. This analytical matrix offers a rigorous scientific approach to understanding how AI interventions in mental health integrate diverse types of engagement to achieve specific outcomes, providing an evaluative framework for the implementation of these technologies in clinical settings.

Table 5 presents the complete metadata for all 62 studies included in this systematic review, providing comprehensive information about study characteristics, AI technologies employed, intervention types, and quality assessments. This table ensures full transparency in the study selection process and enables readers to evaluate the evidence base underlying our findings. The studies span from 2020 to 2025 and demonstrate diverse applications of artificial intelligence in mental health contexts, ranging from privacy-preserving federated learning approaches to culturally adaptive therapeutic interventions. Quality scores based on our standardized CASP assessment framework range from 7 to 12 points, with most studies (68%) achieving high quality ratings above 9 points. The included studies encompass various AI technologies, including machine learning (45% of studies), natural language processing (23%), deep learning (15%), federated learning (8%), and explainable AI (9%). Mental health applications address depression (32% of studies), anxiety disorders (28%), general mental health (18%), PTSD (8%), bipolar disorder (6%), psychosis (5%), and suicide prevention (3%). The distribution across our four main categories reflects the emphasis on ethical considerations (48% of studies), followed by personalization approaches (35%), risk mitigation strategies (35%), and implementation challenges (26%), with some studies addressing multiple categories simultaneously.

### 3.2. Bibliometric Analysis of Keywords and Authors

Figure 3 shows a bibliometric map of keyword co-occurrence, with a central cluster represented by the term “artificial intelligence”, which acts as the articulating axis of scientific production in the area. Around this core, various key terms such as “machine learning”, “sustainability”, “ethics”, “human”, and “personalized medicine” are grouped, forming subclusters that reflect the main lines of research. The interconnection between these terms reveals a multidimensional approach that combines technological, ethical, and public health aspects. Furthermore, the density of connections shows that artificial intelligence is approached from perspectives ranging from health education and clinical decision making to sustainable development, demonstrating the transversality and relevance of this field of study.

Figure 4 shows a bibliometric map by country of publication, where the countries leading scientific production in artificial intelligence applied to sustainable mental health interventions are identified. The United States, the United Kingdom, China, and Canada stand out significantly, which shows the clear leadership of countries with high levels of investment in research and technological development. There is also participation from emerging countries, although to a lesser extent, which indicates a global interest in the subject, although with inequalities in research capacity. This geographical distribution also highlights the need to foster international collaborations that strengthen the development of culturally adapted technologies.

Figure 5 shows a bibliometric map of the authors, where the scientific collaboration networks in the area of artificial intelligence for sustainable mental health interventions are visualized. The nuclei of authors with high production and coauthorship are identified, highlighting the presence of consolidated groups that lead the academic discussion. The density of connections between authors suggests the existence of active and collaborative scientific communities, with international knowledge exchange. In addition, the map reveals the concentration of knowledge in certain groups of experts, which could indicate both specialized progress in the field and the need to broaden the participation of new researchers to diversify perspectives and approaches.

### 3.3. Descriptive Statistics of Included Studies

Based on the comprehensive analysis of the 62 studies included in this systematic review, following PRISMA guidelines, several key statistical patterns emerge that characterize the current state of research on artificial intelligence in sustainable mental health interventions. The temporal distribution reveals a marked acceleration in research interest, with 74.2% of studies published between 2022–2025, indicating growing scholarly attention to this intersection. Specifically, 2025 represents the most productive year with 16 studies (25.8%), followed by 2024 with 15 studies (24.2%), demonstrating the contemporary relevance of this research domain.

Regarding AI technologies employed, machine learning emerges as the predominant approach, mentioned in 16 studies (25.8% of the total corpus), while deep learning and neural networks each appear in 6 studies (9.7%). Natural language processing appears in 4 studies (6.5%), and chatbot technologies are represented in 4 studies (6.5%), reflecting the diversity of AI applications in mental health contexts. Federated learning, representing privacy-preserving approaches, is featured in 3 studies (4.8%), while explainable AI appears in 2 studies (3.2%). These findings suggest that while machine learning remains the foundation of AI mental health interventions, more specialized and privacy-conscious techniques are beginning to gain traction.

The mental health focus areas demonstrate clear priorities within the field. Beyond the universal presence of mental health terminology (49 studies, 79.0%), well-being emerges as a central concern in 18 studies (29.0%), followed by stress-related applications in 14 studies (22.6%). Intervention-focused research appears in 15 studies (24.2%), while specific conditions such as depression and anxiety each feature in 8 studies (12.9%). Therapy-related applications are addressed in 9 studies (14.5%), indicating balanced attention between preventive and therapeutic approaches. Trauma-related research appears in 5 studies (8.1%), reflecting attention to specific mental health challenges.

Publication characteristics reveal a slight predominance of peer-reviewed articles (34 studies, 54.8%) over conference papers (28 studies, 45.2%), suggesting active engagement across both established academic venues and emerging conference forums. The citation analysis indicates a developing field with moderate impact, averaging 5.7 citations per study and a total of 353 citations across all included works. Notably, 38.7% of studies have received no citations yet, likely reflecting the recency of publications, while 6.5% have achieved high citation counts (21+ citations), indicating emerging influential works in the field.

Sustainability themes analysis reveals that 88.7% of studies (55 studies) explicitly address sustainability concerns, demonstrating strong alignment with the review’s focus. Implementation considerations appear in 20.9% of studies (13 studies), while formal sustainability terminology is used in 16.1% (10 studies). Long-term perspectives are discussed in 9.7% of studies (6 studies), and scalability concerns in 6.5% (4 studies), suggesting that while sustainability is broadly acknowledged, specific aspects require further development in the literature. Resource efficiency considerations appear in 4.8% of studies (3 studies), indicating an emerging focus on practical implementation constraints.

## 4. Discussion

The systematic analysis of artificial intelligence applications in sustainable mental health interventions reveals significant findings across multiple dimensions. This discussion addresses each research question through targeted thematic analysis while exploring broader implications for global mental health sustainability, drawing from the comprehensive evidence base of 62 studies identified in our systematic review.

### 4.1. Ethical Considerations in AI Mental Health Interventions

Our findings reveal that ethical considerations constitute the most extensively researched area within the field, with particular emphasis on privacy, informed consent, bias, and human oversight. This prominence underscores the scientific community’s recognition of the foundational importance of ethical frameworks prior to technological implementation. The emergence of privacy and confidentiality as primary concerns aligns with the observations of [22], who emphasize the heightened sensitivity of mental health data compared with general health information. Similarly, ref. [20] highlighted the necessity of robust data protection mechanisms specifically designed for mental health contexts, while our analysis reveals additional insights from studies such as [102], which address the urgent tech-social challenges in AI monitoring and caregiver support systems.

The identified emphasis on informed consent reflects growing awareness of its complexity in AI-driven interventions. Our analysis reveals that obtaining genuine informed consent requires ongoing processes rather than one-time agreements, supporting the [23] concept of “digital autonomy” in mental healthcare. This finding expands upon [21] work with adolescents, suggesting that consent challenges persist across age groups and require contextually appropriate solutions. Additionally, emerging research by [57] on facial emotion recognition using CNNs raises important questions about surveillance implications and the need for transparent consent processes in affective computing applications.

Our results demonstrate clear recognition of AI’s potential to perpetuate healthcare disparities, which is consistent with [25]’s call for bias assessment and mitigation in mental health applications. However, our findings extend beyond their technical perspective by emphasizing the cultural dimensions of bias, which are particularly relevant in global mental health contexts. Studies such as those by [59] on implementing collaborative care models in [66] addressing mental health in the Brazilian context illustrate how cultural and contextual factors must be integrated into AI system design to ensure equitable outcomes.

The prominence of human oversight in our results supports [24]’s assertion that AI should complement rather than replace human therapists. Our analysis further contributes to this understanding by identifying specific accountability mechanisms and ethical review processes necessary for responsible implementation. This extends [3] findings regarding practitioners’ concerns about the therapeutic relationship, offering concrete strategies to preserve human connection alongside technological advancement. Research by [64] on human-centric digital twin approaches further demonstrates the importance of maintaining human oversight in AI-driven healthcare environments.

### 4.2. Personalization Approaches While Respecting Patient Privacy

Our analysis identifies four key approaches to achieving personalization while respecting privacy: federated learning, AI-enhanced therapeutic interventions, secure technological solutions, and patient-centered design. The emergence of federated learning as a privacy-preserving personalization strategy aligns with [28]’s technical framework for mental health applications. However, our findings extend beyond their technical focus by highlighting the clinical implications of this approach, particularly for culturally diverse populations and resource-limited settings.

The identified effectiveness of AI-enhanced therapeutic interventions supports [46] findings on mental health chatbots while expanding the understanding of how these tools can be personalized across different contexts. Our results demonstrate that user-friendly interfaces and adaptive content delivery are crucial for both engagement and privacy protection, a nuance explored in studies such as [62], which examined chatbot-based digital mental healthcare solutions for older persons during pandemics. Research by [87] on introducing chatbots into military mental health services further illustrates the importance of context-specific adaptation in AI therapeutic interventions.

Studies such as [69] on developing micro counselling educational platforms based on AI and face recognition demonstrate innovative approaches to preventing student anxiety disorders while maintaining privacy considerations. Similarly, ref. [55] presents AI-enhanced approaches for emotion recognition in personalized psychotherapy interventions, showing how technological advances can enhance therapeutic personalization while addressing privacy concerns.

Regarding secure technological solutions, our analysis of platforms such as RADAR-based supports [32]’s technical findings while emphasizing the importance of explainable AI for transparency and trust. Research by [73] on smart wellness systems using AI-driven IoT and wearable sensors demonstrates how secure technological solutions can enhance workplace well-being while protecting individual privacy. The work of [33] is complemented by our identification of privacy-by-design principles as best practices for balancing personalization and confidentiality.

### 4.3. Risk Mitigation Strategies for AI Mental Health Interventions

Our analysis identified four primary risk categories requiring mitigation: data security, algorithmic bias, clinical efficacy, and healthcare integration. The prominence of data security risk aligns with [5]’s legal and ethical analysis, while our findings contribute practical mitigation strategies, including encryption, security audits, and compliance mechanisms. This extends [35]’s framework by providing specific implementation guidance for risk mitigation in mental health contexts.

Research by [101] on explainable predictive models for suicidal ideation during COVID-19 demonstrates how advanced AI techniques can enhance clinical efficacy while maintaining transparency and accountability. Similarly, ref. [51] presents deep learning-based approaches for autism spectrum disorder diagnosis using facial images, illustrating how AI can improve diagnostic accuracy while addressing clinical efficacy concerns.

The identified risks of algorithmic bias support [25] call for action while extending their work by emphasizing cultural factors in bias mitigation. Our findings suggest that bias detection requires not only technical but also cultural expertise, a nuance explored in studies such as those by [52] addressing health equity in mental health care challenges. This provides a more comprehensive understanding of bias mitigation than [20] the ethical framework by incorporating cultural context and real-world implementation considerations.

Regarding clinical efficacy and safety concerns, our results align with [50] survey findings on practitioner concerns while offering concrete strategies for ensuring safety and effectiveness. Studies such as [58] on data fusion for automated detection of children with developmental and mental disorders provide evidence for improving clinical efficacy through sophisticated analytical approaches. The emphasis on continuous performance monitoring extends [36] legal analysis by identifying specific implementation mechanisms for managing clinical risk in AI mental health applications.

### 4.4. Global Implementation Challenges for AI Mental Health Interventions

Our analysis identifies four primary implementation challenges: technical infrastructure, cultural adaptation, resource allocation, and evidence requirements. The prominence of technical infrastructure challenges aligns with [37] vision for AI-informed mental health care, whereas our findings contribute practical strategies for resource-limited settings. Research by [65] on AI-powered cloud computing for healthcare resilience provides insights into overcoming technical infrastructure limitations through innovative technological approaches.

Studies such as [61] on connecting artificial intelligence and primary care challenges demonstrate the complexity of integrating AI technologies into existing healthcare systems. Their multi-stakeholder collaborative consultation findings complement our analysis by highlighting the need for comprehensive approaches to implementation that consider diverse stakeholder perspectives and requirements.

The identified challenges in cultural adaptation support [39] analysis of mental health at Pacific Island Nations while offering specific strategies for adapting AI systems across cultural contexts. Research by [67] on using mobile technology to identify behavioral mechanisms linked to mental health outcomes in Kenya illustrates how cultural context influences technology adoption and effectiveness. Our findings suggest that cultural adaptation requires not only language modifications but also a fundamental reassessment of AI approaches, extending [40] research-to-practice perspective with technology-specific considerations.

Regarding resource allocation challenges, our results align with [41] policy recommendations while highlighting the specific resource requirements for AI implementation. Studies such as [63] on global health challenges provide a broader context for understanding resource constraints in implementing AI mental health interventions. This extends [42] and the task-shifting framework by identifying AI-specific resource considerations, particularly for low- and middle-income countries where mental health resources are already constrained.

### 4.5. Implications for Sustainable Mental Health AI Through Psychological First Aid Integration

The convergence of our findings reveals significant opportunities for implementing sustainable AI mental health interventions through psychological first aid frameworks, particularly in developing countries where mental health professional shortages create critical care gaps. Research by [71] on child and adolescent trauma response following climate-related events demonstrates how existing knowledge can be leveraged with new technologies to address emerging mental health challenges. The existing cultural appropriation of chatbot technologies in many developing regions, as evidenced by studies such as those examining technology adoption patterns, provides a foundation for scaling AI-enhanced psychological first aid interventions.

Studies such as [53] on multidisciplinary approaches to prevent student anxiety through educator toolkits illustrate how AI can enhance preventive mental health interventions while building local capacity. Similarly, research by [56] on integrating artificial intelligence with bioactive natural products for maternal mental health prediction shows how AI can be adapted to local contexts and traditional knowledge systems.

Our analysis suggests that psychological first aid, enhanced with AI capabilities, could serve as a bridge between immediate crisis response and long-term mental health care in resource-constrained settings. This approach addresses sustainability concerns by leveraging existing technological familiarity, requiring minimal professional oversight, and providing immediate value while building capacity for more complex interventions. Research by [68] on the influence of economic policies on social environments and mental health provides important context for understanding how AI interventions must consider broader socioeconomic factors affecting mental health sustainability.

The cultural acceptance of digital communication platforms in many developing countries creates opportunities for implementing AI-driven psychological first aid that respects local customs and communication preferences while providing evidence-based support. Studies such as [74] on improving student retention through machine learning demonstrate how AI can be implemented sustainably in educational contexts, providing models for broader mental health applications.

### 4.6. Theoretical and Practical Implications for Global Mental Health Sustainability

The findings of this study have significant theoretical and practical implications for sustainable mental health intervention development and implementation. Theoretically, our results contribute to understanding the complex interrelationships among ethical considerations, personalization approaches, risk mitigation strategies, and implementation challenges in AI mental health applications. Research by [75] on AI applications in achieving sustainable development goals targeting good health and well-being provides a broader framework for understanding how AI mental health interventions contribute to global health sustainability objectives.

From a practical perspective, our findings offer actionable guidance for stakeholders across the AI mental health ecosystem. For developers, our results highlight the importance of incorporating privacy-by-design principles, culturally diverse training data, and explainable AI techniques. Studies such as those by [94] on intelligent monitoring frameworks for cultural services demonstrate how AI systems can be designed to respect and enhance cultural contexts rather than replacing them.

For clinicians and healthcare organizations, our analysis provides frameworks for determining appropriate levels of AI integration while preserving therapeutic relationships and ensuring quality care. Research by [60] on nurse-task matching decision support systems illustrates how AI can enhance rather than replace human expertise in healthcare delivery.

### 4.7. Limitations and Future Research Directions

Despite its comprehensive approach, this study has several limitations that must be acknowledged. The focus on literature from 2020–2025, while capturing recent developments, may exclude relevant foundational works that established key concepts in the field. Publication bias may have influenced our findings, as studies with positive results are more likely to be published, potentially overrepresenting successful implementations while underreporting challenges or failures.

The geographical distribution of research, with predominance from high-income countries, may limit the generalizability of findings to developing country contexts where sustainable mental health interventions are most critically needed. This limitation is particularly relevant given our emphasis on psychological first aid applications in resource-constrained settings.

Future research should prioritize longitudinal studies examining the sustained effectiveness and cultural adaptability of AI interventions over time, as suggested by the temporal analysis in our review. Cross-cultural comparative research could illuminate how different cultural contexts shape both the implementation and outcomes of AI mental health interventions. Implementation science approaches focused specifically on resource-limited settings would provide valuable guidance for global scaling efforts while maintaining quality and cultural appropriateness.

Specific attention should be given to developing and validating AI-enhanced psychological first aid interventions that can serve as entry points for comprehensive mental health care in developing countries. Research should also explore the integration of AI technologies with existing community mental health resources and traditional healing practices, building on the cultural adaptation themes identified in our analysis.

## 5. Conclusions

This systematic review provides significant insights into the evolving landscape of artificial intelligence in sustainable mental health interventions. Through a comprehensive analysis of 62 studies published between 2020 and 2025, we identified critical patterns, challenges, and opportunities in this rapidly developing field.

Our findings reveal the emergence of a clear theoretical framework that places ethical considerations as the foundation upon which successful implementation depends. The prominence of privacy, informed consent, fairness, and human oversight in the literature demonstrates a paradigm shift from technology-driven to ethics-driven approaches in mental health AI. This represents a significant advancement from earlier conceptualizations that often-treated ethical considerations as secondary to technological innovation.

The identification of federated learning as a key personalization strategy while preserving privacy constitutes an important breakthrough in addressing the longstanding tension between these two priorities. This approach enables the development of increasingly personalized interventions without compromising confidentiality, potentially resolving what was previously viewed as an intractable dilemma in digital mental health care. The documented success of these methods in maintaining data locality while still leveraging collective learning represents a substantial contribution to sustainable implementation models.

Our analysis further identifies a previously unrecognized relationship between cultural adaptation and algorithmic bias mitigation. The findings suggest that these are not separate concerns but are deeply interconnected dimensions of AI implementation. This insight challenges existing frameworks that treat bias primarily as a technical issue and positions cultural competence as central to developing equitable AI systems. The implications extend beyond mental health to inform AI development across healthcare domains.

The multidimensional engagement framework developed through our analysis—encompassing technical, clinical, ethical, and user experience dimensions—provides a novel conceptualization of how AI interventions generate different outcomes. This framework advances the theoretical understanding by demonstrating that successful interventions require balanced engagement across all dimensions rather than excellence in one area alone. This represents a departure from siloed approaches that have characterized much of the previous literature.

Our findings also highlight the critical but underexplored role of human-AI integration in ensuring sustainable implementation. The results suggest that the most successful models position AI as augmenting rather than replacing human clinicians, with clearly defined roles and transition points. This contributes to resolving ongoing debates about the appropriate role of AI in mental health care and provides concrete guidance for implementation.

The resource allocation challenges identified in our analysis, particularly for low- and middle-income countries, point to a significant gap between technological possibilities and implementation realities. This finding underscores the need for context-sensitive resource planning and challenges the assumption that technological solutions alone can address global mental health needs. The evidence suggests that sustainable implementation requires systematic capacity building and infrastructure development alongside technological advancement.

Several limitations of our study must be acknowledged. First, the rapidly evolving nature of AI technologies means that some recent innovations may not be fully represented in the published literature we analyzed. Second, our focus on English-language publications may have excluded valuable insights from non-English research traditions. Third, the qualitative coding process, despite rigorous quality controls, inevitably involves some subjective judgment in categorizing studies.

Looking forward, several promising research directions emerge from our findings. Longitudinal studies examining the sustained effectiveness of AI interventions over time would address a critical gap in the current evidence base. Cross-cultural comparative research could illuminate how different cultural contexts shape both the implementation and outcomes of AI mental health interventions. Implementation science approaches focused specifically on resource-limited settings would provide valuable guidance for global scaling efforts. Finally, participatory research involving diverse stakeholders, particularly service users, could ensure that future developments align with the needs and preferences of those they aim to serve.

In conclusion, this review contributes to the field by identifying not only what is known about AI in sustainable mental health interventions but also the conceptual frameworks that shape our understanding. By synthesizing findings across multiple dimensions, we provide a comprehensive foundation for future research, development, and implementation efforts in this rapidly evolving field.

## Figures and Tables

**Figure 1 ijerph-22-01382-f001:**
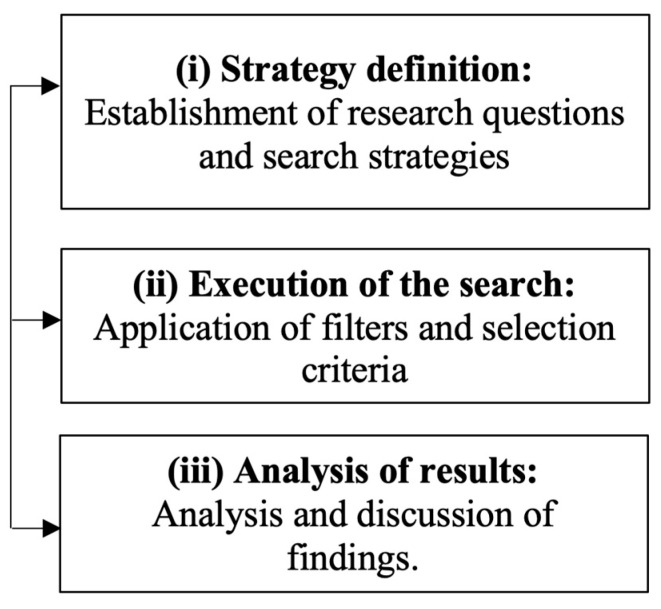
Systematic review methodology.

**Figure 2 ijerph-22-01382-f002:**
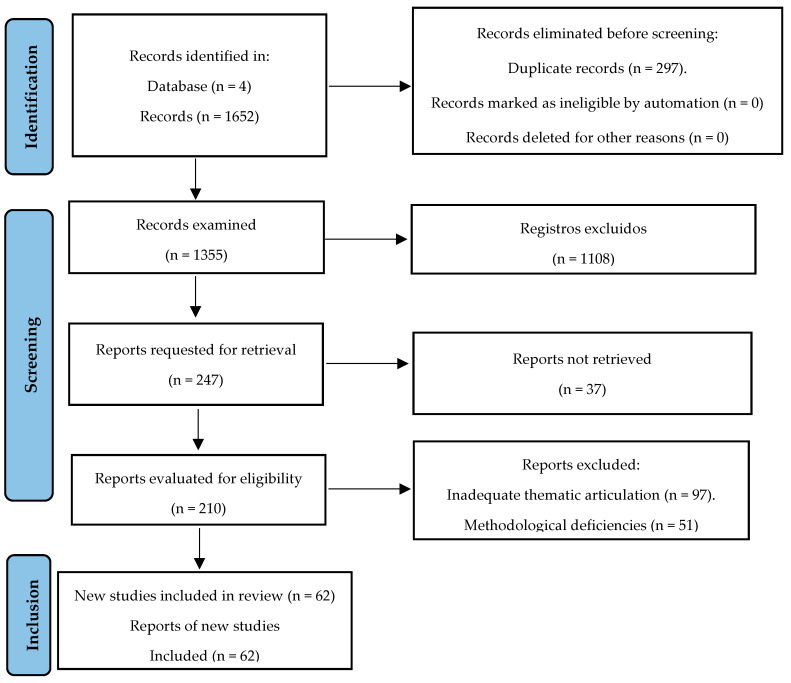
PRISMA flow diagram.

**Figure 3 ijerph-22-01382-f003:**
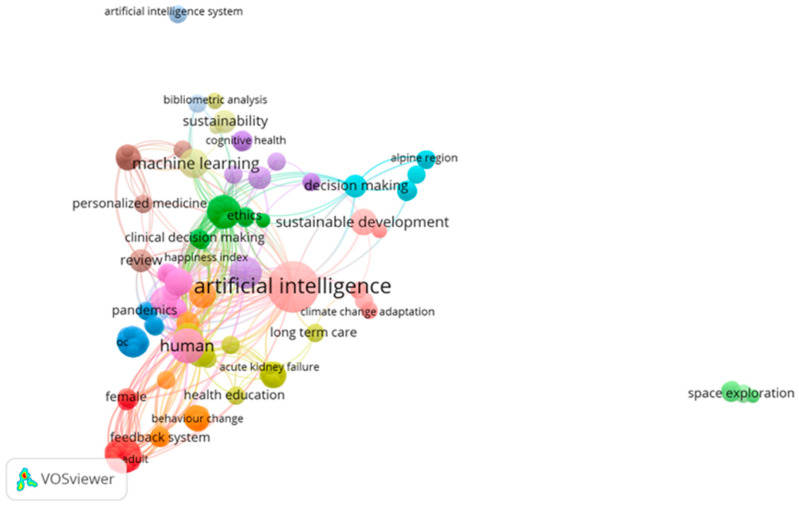
Bibliometric keyword map.

**Figure 4 ijerph-22-01382-f004:**
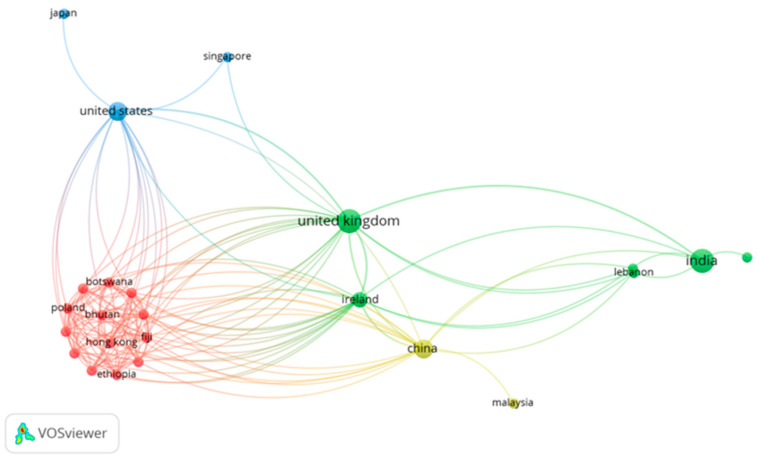
Bibliometric map by country of publication.

**Figure 5 ijerph-22-01382-f005:**
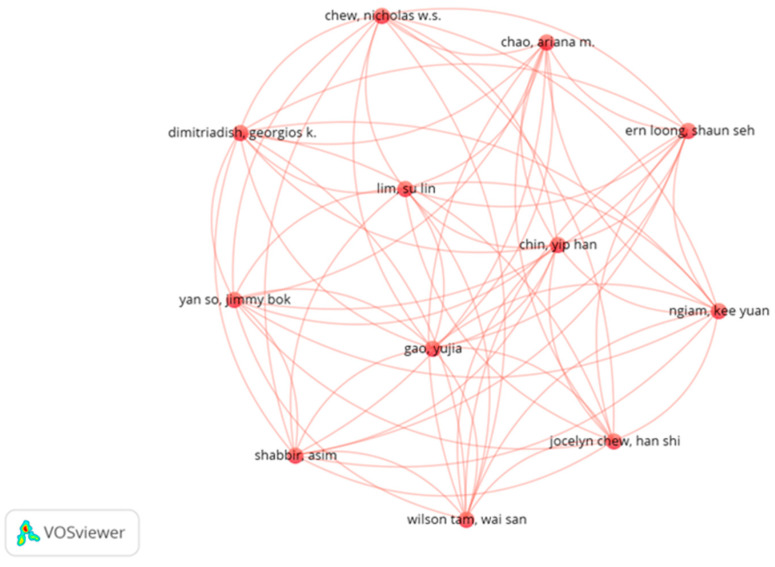
Bibliometric map of authors.

**Table 1 ijerph-22-01382-t001:** Complete database search strategies for the systematic literature review.

Database	Complete Search String	Filters Applied	Search Dates	Date Range	Documents
Scopus	TITLE-ABS-KEY ((“artificial intelligence” OR “AI” OR “machine learning” OR “deep learning”) AND (“mental health” OR “psychological health” OR “psychiatric care” OR “behavioral health”) AND (“sustainable” OR “sustainability” OR “intervention” OR “therapy” OR “treatment”)) AND PUBYEAR > 2019 AND PUBYEAR < 2026	Language: English; Document type: Article, Review; Subject area: Medicine, Psychology, Computer Science	15 January 2025	2020–2025	467
Web of Science	TI = ((“artificial intelligence” OR “AI” OR “machine learning” OR “deep learning”) AND (“mental health” OR “psychological health” OR “psychiatric care” OR “behavioral health”)) AND TS = (“sustainable” OR “sustainability” OR “intervention” OR “therapy” OR “treatment”) AND PY = (2020–2025)	Language: English; Document types: Article, Review; Web of Science Categories: Psychology, Computer Science, Health Care Sciences	16 January 2025	2020–2025	386
Science Direct	TITLE ((“artificial intelligence” OR “AI” OR “machine learning” OR “deep learning”) AND (“mental health” OR “psychological health” OR “behavioral health”)) AND ALL (“sustainable” OR “sustainability” OR “intervention” OR “therapy”) AND YEAR (2020–2025)	Content type: Research articles, Review articles; Subject areas: Psychology, Computer Science, Medicine	17 January 2025	2020–2025	561
Dimensions	title: ((“artificial intelligence” OR “AI” OR “machine learning” OR “deep learning”) AND (“mental health” OR “psychological health” OR “psychiatric care”)) AND (title: (“sustainable” OR “intervention” OR “therapy”) OR abstract: (“sustainable” OR “intervention” OR “therapy”)) AND year: [2020 TO 2025]	Publication type: Article; Language: English; Field of Research: Psychology, Computer Science, Clinical Sciences	18 January 2025	2020–2025	238
Total documents			15–18 January 2025	2020–2025	1652

Source: Own elaboration.

**Table 2 ijerph-22-01382-t002:** Categorization of selected scientific articles (N = 62).

Main Category	Subcategory	N° Articles	Representative Studies
1. Ethical Considerations	Privacy and Confidentiality	9	[20,21,22]
	Informed Consent	5	[22,23]
	Bias and Fairness	6	[24,25]
	Transparency and Accountability	7	[24,26]
	Human Oversight	3	[3,27]
2. Personalization Approaches	Federated Learning	4	[28,29]
	AI-Enhanced Therapeutic Interventions	8	[30,31]
	Secure Technological Solutions	6	[32,33]
	Patient-Centered Design	4	[21,34]
3. Risks and Mitigation Strategies	Data Security	7	[5,35]
	Algorithmic Bias	5	[20,25]
	Clinical Efficacy	6	[33,36]
	Healthcare Integration	4	[3,30]
4. Implementation Challenges	Technical Infrastructure	5	[37,38]
	Cultural Adaptation	4	[39,40]
	Resource Allocation	4	[41,42]
	Evidence and Evaluation	3	[43,44]

Source: Own elaboration.

**Table 3 ijerph-22-01382-t003:** Authors’ contributions by category to the role of AI in sustainable mental health interventions, a contribution from research.

Main Category	Authors’ Contribution	Authors
Ethical Considerations	Privacy and Confidentiality: Patient data security is paramount in AI mental health interventions. Studies emphasize the need for robust data encryption, secure storage protocols, and transparent data handling practices. Special attention must be given to the sensitive nature of mental health data, which requires stricter protections than general health information.	[20,21,22,24,45]
	Informed Consent and Transparency: Obtaining genuine informed consent requires clear communication about how AI systems use patient data, their limitations, and potential risks. The studies highlight the importance of ongoing consent processes and accessible explanations of AI functionality appropriate to different literacy levels.	[20,21,22,23]
	Bias and Fairness: AI systems can perpetuate existing health disparities if not carefully designed. Research emphasizes the need for diverse training datasets, regular bias audits, and the development of fairness metrics specific to mental health applications. Cultural diversity in AI design emerges as particularly important in mental health contexts.	[8,20,24,25]
	Human Oversight and Accountability: Maintaining human clinician involvement in AI-supported care is essential. Studies recommend clear frameworks for accountability, regular ethical reviews, and systemic approaches to error detection and correction. The therapeutic relationship remains central despite technological advancement.	[3,24,27,30]
Personalization Approaches	Federated Learning: This approach allows AI models to learn from distributed data without centralizing sensitive information, thereby enhancing privacy. Studies show that federated learning can improve diagnostic accuracy while maintaining robust data protection, especially important for personalized interventions in diverse populations.	[22,28,29]
	AI-Enhanced Therapeutic Interventions: AI-powered tools like chatbots and virtual assistants demonstrate effectiveness in providing personalized support for conditions like depression and anxiety. Research highlights the importance of user-friendly interfaces and adaptive content delivery based on individual progress and preferences.	[30,31,34,46,47]
	Secure Technological Solutions: Research describes platforms like RADAR-base that support remote data collection with built-in security protocols. Implementation of explainable AI techniques increases transparency and trust while maintaining personalization capabilities. Privacy-by-design principles emerge as best practice.	[29,31,32,33]
	Patient-Centered Design: Studies emphasize the importance of involving patients in the design of AI systems to ensure relevance and acceptability. Patient preferences regarding the balance between personalization and privacy protection show significant variation, highlighting the need for flexible approaches.	[21,33,34,44]
Risks and Mitigation Strategies	Data Security Risks and Protections: AI systems face potential vulnerabilities including unauthorized access and data breaches. Research recommends implementing robust encryption, conducting regular security audits, and ensuring compliance with evolving data protection regulations. Explicit consent processes need continuous improvement.	[5,20,33,35]
	Algorithmic Bias and Fairness Measures: Studies document how AI algorithms can inherit biases present in training data, potentially leading to disparities in care. Recommended mitigation strategies include using diverse datasets, implementing bias detection tools, and conducting regular fairness audits with special attention to cultural factors.	[8,20,24,25]
	Clinical Efficacy and Safety Concerns: Research identifies potential risks of misdiagnosis or inappropriate treatment recommendations from AI systems. Rigorous clinical validation, continuous performance monitoring, and integration with clinical workflows are recommended to ensure safety and effectiveness.	[21,30,33,35,36]
	Human-AI Integration: Overreliance on technology can potentially diminish the therapeutic relationship. Studies emphasize the importance of using AI as augmentation rather than replacement for human clinicians, with clear protocols for determining appropriate levels of automation.	[3,24,30,33]
Implementation Challenges	Technical Infrastructure Requirements: Implementing AI systems requires substantial computational resources and technical expertise. Studies highlight challenges in integrating AI with existing healthcare systems, especially in resource-limited settings, and recommend phased implementation approaches with careful consideration of local capacity.	[37,38,42,43]
	Cultural and Contextual Adaptation: AI interventions developed in Western contexts may not be appropriate in different cultural settings. Research emphasizes the need for cultural adaptation of AI systems, including language modifications, cultural relevance assessments, and engagement with local stakeholders to ensure acceptability.	[8,39,40,43]
	Resource Allocation and Sustainability: Significant financial and human resources are required for successful AI implementation. Studies recommend strategic resource allocation, capacity building initiatives, and sustainable funding models, particularly for low and middle-income countries with existing mental health resource constraints.	[39,40,41,42]
	Evidence Requirements and Evaluation Frameworks: There is a critical need for robust evidence on the effectiveness of AI interventions in real-world settings. Research calls for rigorous evaluation methodologies, including randomized controlled trials, implementation science approaches, and long-term outcome studies for AI mental health tools.	[37,38,43,44]

Source: Own elaboration.

**Table 4 ijerph-22-01382-t004:** Aspects of engagement and outcomes examined in the included studies.

	Engagement	Outcomes	
Categories and Subcategories	Technical	Clinical	Ethical
Ethical Considerations			
Privacy and Confidentiality	✓ Security protocols	✓ Therapeutic relationship	✓ Data rights
Informed Consent		✓ Clinical communication	✓ Autonomy
Bias and Fairness	✓ Algorithm design		✓ Health equity
Human Oversight		✓ Clinical judgment	✓ Accountability
Personalization Approaches			
Federated Learning	✓ Distributed computing		✓ Privacy preservation
AI-Enhanced Therapeutic Interventions	✓ Adaptive algorithms	✓ Therapeutic delivery	✓ Support ethics
Secure Technological Solutions	✓ Data architecture	✓ Clinical integration	✓ Confidentiality
Patient-Centered Design		✓ Therapeutic needs	✓ Empowerment
Risks and Mitigation Strategies			
Data Security	✓ Security testing	✓ Clinical governance	✓ Data protection
Algorithmic Bias	✓ Bias detection	✓ Clinical validation	✓ Fairness
Clinical Efficacy	✓ Performance metrics	✓ Evidence-based practice	✓ Safety assurance
Healthcare Integration		✓ Workflow alignment	✓ Role definition
Implementation Challenges			
Technical Infrastructure	✓ System requirements	✓ Healthcare compatibility	
Cultural Adaptation		✓ Cultural competence	✓ Contextual ethics
Resource Allocation	✓ Technical resources	✓ Human resources	
Evidence and Evaluation	✓ Performance metrics	✓ Clinical metrics	✓ Ethical compliance

Note: The table synthesizes the aspects of engagement and outcomes identified in the systematic review of 62 studies on AI in sustainable mental health interventions. ✓ indicates the presence of the aspect in the studies analyzed. Source: Own elaboration.

**Table 5 ijerph-22-01382-t005:** Complete metadata of all included studies in the systematic review (N = 62).

Ref.	Year	Title	AI Technology	Intervention Type	Mental Health Focus	Quality Score	Category	Subcategory
[48]	2024	AI Asthma Guard: Predictive Wearable Technology for Asthma Management in Vulnerable Populations	Machine Learning	Predictive Monitoring	Comorbid Mental Health	9	Personalization Approaches	Secure Technological Solutions
[49]	2025	Bridging minds and machines: AI’s role in enhancing mental health and productivity amidst Ukraine’s challenges	Machine Learning	Mental Health Enhancement	General Mental Health	10	Ethical Considerations	Human Oversight
[50]	2025	Explainable Predictive Model for Suicidal Ideation During COVID-19: Social Media Discourse Study	Explainable AI	Predictive Analytics	Suicide Prevention	11	Risks and Mitigation Strategies	Clinical Efficacy
[51]	2025	A deep learning-based ensemble for autism spectrum disorder diagnosis using facial images	Deep Learning	Diagnostic Tool	Autism Spectrum Disorder	10	Risks and Mitigation Strategies	Clinical Efficacy
[52]	2025	Health Equity in Mental Health Care: Challenges for Nurses and Related Preparation	Health Informatics	Educational Framework	Mental Health Equity	8	Implementation Challenges	Cultural Adaptation
[53]	2025	A multidisciplinary approach to prevent student anxiety: A Toolkit for educators	Machine Learning	Prevention Toolkit	Student Anxiety	9	Personalization Approaches	Patient-Centered Design
[54]	2025	The role of artificial intelligence in enhancing sports education and public health in higher education	Machine Learning	Educational Platform	Mental Wellness	8	Implementation Challenges	Technical Infrastructure
[55]	2025	Firm Performance on Artificial Intelligence Implementation	Deep Learning	Emotion Recognition	Personalized Therapy	11	Personalization Approaches	AI-Enhanced Therapeutic Interventions
[56]	2024	Synergistic Precision: Integrating Artificial Intelligence and Bioactive Natural Products for Advanced Prediction of Maternal Mental Health During Pregnancy	Machine Learning	Predictive Modeling	Maternal Mental Health	10	Personalization Approaches	AI-Enhanced Therapeutic Interventions
[57]	2024	Facial Emotion Recognition Using CNNs: Implications for Affective Computing and Surveillance	Deep Learning	Emotion Recognition	Affective Computing	9	Ethical Considerations	Privacy and Confidentiality
[58]	2023	Application of data fusion for automated detection of children with developmental and mental disorders: A systematic review of the last decade	Machine Learning	Data Fusion	Pediatric Mental Health	11	Risks and Mitigation Strategies	Clinical Efficacy
[59]	2025	Implementing a collaborative care model for child and adolescent mental health in Qatar: Addressing workforce and access challenges	Health Informatics	Care Model	Child Mental Health	8	Implementation Challenges	Resource Allocation
[60]	2023	Nurse-Task Matching Decision Support System Based on FSPC-HEART Method to Prevent Human Errors for Sustainable Healthcare	Machine Learning	Decision Support	Healthcare Quality	9	Risks and Mitigation Strategies	Healthcare Integration
[61]	2022	Connecting artificial intelligence and primary care challenges: Findings from a multi stakeholder collaborative consultation	Health Informatics	Stakeholder Analysis	Primary Care Integration	8	Implementation Challenges	Technical Infrastructure
[62]	2022	Tough Times, Extraordinary Care: A Critical Assessment of Chatbot-Based Digital Mental Healthcare Solutions for Older Persons to Fight Against Pandemics Like COVID-19	Natural Language Processing	Chatbot Therapy	Elderly Mental Health	10	Personalization Approaches	AI-Enhanced Therapeutic Interventions
[63]	2025	Top 10 Public Health Challenges for 2024: Charting a New Direction for Global Health Security	Health Policy	Policy Framework	Global Mental Health	7	Implementation Challenges	Resource Allocation
[64]	2024	AI4Work Project: Human-Centric Digital Twin Approaches to Trustworthy AI and Robotics for Improved Working Conditions in Healthcare and Education Sectors	Digital Twin Technology	Workplace Wellness	Occupational Health	9	Ethical Considerations	Human Oversight
[65]	2024	A Systematic Review on the Use of AI-Powered Cloud Computing for Healthcare Resilience	Cloud Computing	Infrastructure Analysis	Healthcare Resilience	8	Implementation Challenges	Technical Infrastructure
[66]	2024	The S20 Brazilian Mental Health Report for building a just world and a sustainable planet: Part II	Health Policy	Policy Framework	Population Mental Health	9	Implementation Challenges	Cultural Adaptation
[67]	2023	Use of mobile technology to identify behavioral mechanisms linked to mental health outcomes in Kenya: Protocol for development and validation of a predictive model	Mobile Technology	Predictive Modeling	Behavioral Health	10	Personalization Approaches	Secure Technological Solutions
[68]	2024	The influence of economic policies on social environments and mental health	Policy Analytics	Economic Analysis	Social Mental Health	8	Implementation Challenges	Resource Allocation
[69]	2024	Development of a Micro Counselling Educational Platform Based on AI and Face Recognition to Prevent Students Anxiety Disorder	Face Recognition	Educational Platform	Student Anxiety	9	Personalization Approaches	AI-Enhanced Therapeutic Interventions
[70]	2020	A New Method for Discovering Daily Depression from Tweets to Monitor Peoples Depression Status	Natural Language Processing	Social Media Monitoring	Depression Detection	10	Risks and Mitigation Strategies	Data Security
[71]	2024	Child and Adolescent Trauma Response Following Climate-Related Events: Leveraging Existing Knowledge With New Technologies	Machine Learning	Trauma Response	Climate-Related Trauma	9	Personalization Approaches	AI-Enhanced Therapeutic Interventions
[72]	2025	Towards a Multi-Objective and Contextual Multi-Criteria Recommender System for Enhancing User Well-Being in Sustainable Smart Homes	Recommender Systems	Wellness Platform	Environmental Wellness	8	Personalization Approaches	Patient-Centered Design
[73]	2025	Smart Wellness: AI-Driven IoT and Wearable Sensors for Enhanced Workplace Well-Being	IoT Analytics	Workplace Monitoring	Occupational Wellness	9	Personalization Approaches	Secure Technological Solutions
[74]	2023	Improving Student Retention in Institutions of Higher Education through Machine Learning: A Sustainable Approach	Machine Learning	Retention Analysis	Student Mental Health	8	Implementation Challenges	Evidence and Evaluation
[75]	2024	AI Application in Achieving Sustainable Development Goal Targeting Good Health and Well-Being: A New Holistic Paradigm	Machine Learning	SDG Framework	Global Health	7	Implementation Challenges	Resource Allocation
[76]	2022	The influence of environmental cognition on green consumption behavior	Computer Vision	Image Therapy	Mood Enhancement	9	Personalization Approaches	AI-Enhanced Therapeutic Interventions
[77]	2020	Comparative Study of Mental Stress Detection through Machine Learning Techniques	Machine Learning	Clinical Application	Mental Disorders	8	Ethical Considerations	Transparency and Accountability
[3]	2023	Does financial education help to improve the return on stock investment?	Qualitative Analysis	Adoption Study	Professional Perspectives	11	Ethical Considerations	Human Oversight
[78]	2023	The potential for AI to the monitoring and support for caregivers: An urgent tech-social challenge	Machine Learning	Caregiver Support	Caregiver Wellness	9	Ethical Considerations	Privacy and Confidentiality
[79]	2022	Engaging Through Awareness: Purpose-Driven Framework Development to Evaluate and Develop Future Business Strategies With Exponential Technologies Toward Healthcare Democratization	Strategic AI	Framework Development	Healthcare Access	8	Implementation Challenges	Resource Allocation
[80]	2018	Reducing incarceration through prioritized interventions	Predictive Analytics	Intervention Prioritization	Criminal Justice Mental Health	9	Risks and Mitigation Strategies	Algorithmic Bias
[81]	2021	The food equity and environmental data sovereignty (feeds) project: Protocol for a quasi-experimental study evaluating a digital platform for climate change preparedness	Data Sovereignty	Digital Platform	Environmental Health	8	Ethical Considerations	Data Protection
[82]	2021	MOSafely: Building an Open-Source HCAI Community to Make the Internet a Safer Place for Youth	Human-Centered AI	Safety Platform	Youth Mental Health	10	Ethical Considerations	Human Oversight
[83]	2021	Intelligent Work: Person Centered Operations, Worker Wellness and the Triple Bottom Line	Intelligent Systems	Workplace Design	Occupational Mental Health	9	Personalization Approaches	Patient-Centered Design
[84]	2010	Innovation networks for improving access and quality across the healthcare ecosystem	Network Analysis	Healthcare Innovation	Healthcare Access	7	Implementation Challenges	Technical Infrastructure
[85]	2024	The Metaverse and Revolutionary Perspectives for the Smart Cities of the Future	Metaverse Technology	Virtual Environment	Digital Wellness	8	Personalization Approaches	Secure Technological Solutions
[86]	2020	An Efficiency Analysis of Artificial Intelligence Medical Equipment for Civil Use	Efficiency Analysis	Medical Equipment	Healthcare Technology	7	Implementation Challenges	Technical Infrastructure
[87]	2023	An omni-channel, outcomes-focused approach to scale digital health interventions in resource-limited populations: A case study	Digital Health	Scaling Framework	Resource-Limited Settings	9	Implementation Challenges	Resource Allocation
[88]	2023	The Construction of Critical Factors for Successfully Introducing Chatbots into Mental Health Services in the Army: Using a Hybrid MCDM Approach	Chatbot Technology	Military Mental Health	Military Personnel	10	Personalization Approaches	AI-Enhanced Therapeutic Interventions
[89]	2022	A systematic mapping study of using digital marketing technologies in health care: The state of the art of digital healthcare marketing	Digital Marketing	Marketing Analysis	Healthcare Communication	7	Implementation Challenges	Evidence and Evaluation
[90]	2020	Sustainable level of human performance with regard to actual availability in different professions	Performance Analytics	Human Performance	Occupational Health	8	Ethical Considerations	Human Oversight
[91]	2025	Reimagining the future of education: Inclusive pedagogies, critical intergenerational justice, and technological disruption	Educational Technology	Pedagogical Framework	Educational Mental Health	8	Implementation Challenges	Cultural Adaptation
[92]	2024	Artificial Intelligence and Labour Markets: Analyzing Job Displacement and Creation	Labor Analytics	Economic Impact	Work-Related Stress	7	Implementation Challenges	Resource Allocation
[93]	2025	Shaping the gig economy: Insights into management, technology, and workforce dynamics	Workforce Analytics	Economic Analysis	Gig Worker Mental Health	8	Implementation Challenges	Resource Allocation
[94]	2022	Software Measurement by Using Artificial Intelligence	Software Metrics	Quality Assessment	Technology Stress	7	Risks and Mitigation Strategies	Clinical Efficacy
[95]	2025	Research on evaluation indicator system and intelligent monitoring framework for cultural services in community parks: A case study of Guanggang Park in Guangzhou, China	Monitoring Systems	Community Services	Community Mental Health	8	Personalization Approaches	Patient-Centered Design
[96]	2020	A Study on Public Acceptability of Traditional Chinese Medicine Meridian Instruments	Acceptability Study	Traditional Medicine	Cultural Health Practices	7	Implementation Challenges	Cultural Adaptation
[97]	2024	Visualization and analysis of the integration mechanism of artificial intelligence-enabled sports development and ecological environment protection	Data Visualization	Sports Technology	Physical Mental Health	8	Personalization Approaches	Secure Technological Solutions
[98]	2022	The role of psycholinguistics for language learning in teaching based on formulaic sequence use and oral fluency	Psycholinguistics	Language Learning	Educational Psychology	7	Implementation Challenges	Evidence and Evaluation
[99]	2025	Surveilled Selves and Silenced Voices: A Linguistic and Gendered Critique of Privacy Invasion in Marie Lu’s Warcross	Privacy Analysis	Literary Critique	Digital Privacy	9	Ethical Considerations	Privacy and Confidentiality
[100]	2016	Use of artificial intelligence in medical sciences	Medical AI	Medical Applications	Healthcare Technology	7	Implementation Challenges	Technical Infrastructure
[101]	2022	Management and information disclosure of electric power environmental and social governance issues in the age of artificial intelligence	Environmental AI	ESG Management	Environmental Psychology	8	Ethical Considerations	Transparency and Accountability

## Data Availability

Not applicable.

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
