# Peer review of "Sustainability of AI-Assisted Mental Health Intervention: A Review of the Literature from 2020–2025"

_ijerph, 2025, doi:10.3390/ijerph22091382_

Round 1

Reviewer 1 Report (Previous Reviewer 1)

Comments and Suggestions for Authors

I would like to thank the authors for their comprehensive and detailed revisions of the manuscript titled "The Role of Artificial Intelligence in Sustainable Mental Health Interventions: A Review of the Literature from 2020–2025." The authors have responded constructively to all reviewer feedback and presented significant improvements that increase the transparency, methodological rigor, and conceptual coherence of the manuscript.

As Reviewer 1, I highlight the following major improvements:

  • Authors have now provided a full table (Table 5) of all 62 studies included with study-level metadata such as year of publication, type of AI, type of intervention, and country, thus addressing study-level transparency concerns.
  • The search strategy description with Boolean logic, database filters, and search dates has been provided, enhancing the reproducibility of the review.
    • Authors now report inter-rater agreement (Cohen's κ = 0.78), and CASP is applied for quality evaluation, providing immense authority to the review process.
    • Descriptive statistics are provided to complement thematic findings (e.g., % of studies by AI type and mental health focus), improving on the former lack of quantification.
    • Bias issues, such as publication, language, and geographic bias, are now mentioned and discussed.
    • The argument has been simplified with more conceptual depth and clarity. Of particular interest is the introduction of psychological first aid as a model of AI-based mental health care in low-income countries, which fits extremely well with the manuscript's emphasis on sustainability.
    • Bibliometric analysis with VOSviewer and new visualizations now offer more analytical traction and context.

  • Practical intervention concerns of Reviewer 2 were addressed by a more conceptual exploration of AI as a mediator of care provision.

  • Terminology concerns, paragraph consistency, and sustainability dimension justification concerns of Reviewer 3 were addressed accordingly.

All of the above major and minor issues pointed out by the reviewers have been duly considered. The updated manuscript is now publishable in IJERPH, as a timely and methodologically rigorous contribution to AI, mental health, and sustainability scholarship.

Author Response

Answer: Dear reviewer, thank you for your assessment of our work.

Reviewer 2 Report (Previous Reviewer 3)

Comments and Suggestions for Authors

The authors have extensively revised the manuscript. Yet the mismatch between the title and the actual content is still quite noticeable. In particular, the term "sustainable mental health interventions" in the title is ambiguous and problematic. The authors added some discussion of AI sustainability in the introduction, but this does not address the core issue: the sustainability of "mental health interventions", rather than the sustainability of AI itself. (A lot of newly added texts are discussing the sustainability of AI-assisted mental health intervention) Moreover, the final categories identified through the systematic review focuses mainly on potential risks of using AI in mental health, rather than providing any insight into intervention sustainability.

My suggestion is that the authors should more clearly define what would constitute a sustainable mental health intervention (and this requires a justification for raising this issue in the first place; e.g., are there concerns about mental health intervention being not sustainable?), then analyze how the use of AI would be detrimental to or beneficial for the sustainability of mental health intervention. Otherwise, despite the painstaking work of review, the manuscript does not fully deliver what its title suggest that it offers. Another possibility would simply be to change the title to the likes of “sustainability of AI-assisted mental health intervention”.

Author Response

Answer: Good point, dear reviewer. Since we don't want to complicate things and for the sake of rationality, we will modify the title as you suggested.

Round 2

Reviewer 2 Report (Previous Reviewer 3)

Comments and Suggestions for Authors

The authors have revised the manuscript. I have no further comments.

This manuscript is a resubmission of an earlier submission. The following is a list of the peer review reports and author responses from that submission.

Round 1

Reviewer 1 Report

Comments and Suggestions for Authors

This manuscript is a systematic review of 62 peer-reviewed papers issued between 2020 and 2025 identifying how artificial intelligence (AI) can support sustainable mental health intervention. The authors use PRISMA guidelines and combine bibliometric analysis with thematic synthesis, divided across four dimensions: ethical concerns, personalization approaches, managing risk, and implementation challenges. The review is focused on addressing relevant and modern research topics, situates findings within existing literature, and returns useful implications to clinical, technical, and policy stakeholders.

  • Insufficient Study-Level Summary: No appendix or table presents a list of all 62 included studies with details such as year, methodology, and country.
  • Missing Search String and Timeframe Information: There is no complete database searching and particular search dates available, making it less reproducible.
  • Restrictive Use of Descriptive Statistics: The thematic synthesis is qualitative in nature, yet simple numerical summary (i.e., frequency of subthemes) would be valuable.
  • Inter-Rater Reliability Not Provided: Even though a dual-review process is in place, inter-rater agreement statistics like Cohen's κ are not reported.
  • Bias Across Studies Not Comprehensive: While the quality of the individual studies is assessed, publication or language bias is discussed barely at all.

    Major Concerns
  • Study Selection Transparency and Data Extraction Supply a comprehensive table or appendix of all included studies and their metadata (e.g., uthors, year, type of AI, intervention type, quality rating).
  • Record full database search strings (e.g., Boolean logic, filters) and search date ranges for all four databases.
  • Reporting of Inter-Rater Reliability State study inclusion and CASP scoring agreement statistics (e.g., percentage agreement or Cohen's κ) to increase methodological transparency.
  • Address Risk of Bias Across Studies Describe how publication bias, funding bias, or language bias could have influenced the evidence base.

Minor Suggestions

  • Define Thematic Categories: Enforce more accurate subheadings in the discussion and results such that each topic corresponds directly to research questions.
  • Sharpen Assertions of Novelty: Avoid using phrases like "previously unrecognized" unless fully justified. "Underexplored" is perhaps preferable.
  • Report Descriptive Statistics: Attempt to report simple statistics such as the percentage of studies that address each subtheme or use certain AI techniques.
  • Improve Table Reference: Properly cite tables and figures with consecutively numbered references throughout the paper.

The article is responding to a modern and current question through appropriate organization and coherence. Transparency, reproducibility, and consistency of reporting need to be improved, however, if a systematic review needs to be to full standards.

Author Response

This manuscript is a systematic review of 62 peer-reviewed articles, published between 2020 and 2025, that identifies how artificial intelligence (AI) can support sustainable mental health intervention. The authors use PRISMA guidelines and combine bibliometric analysis with thematic synthesis, divided into four dimensions: ethical concerns, personalization approaches, risk management, and implementation challenges. The review focuses on addressing relevant and contemporary research topics, contextualizes findings within existing literature, and offers useful implications for clinical, technical, and policy stakeholders.

  • Insufficient study-level summary: no appendix or table presents a list of the 62 included studies with details such as year, methodology, and country.
  • Missing information about search string and time period: no complete database search or specific search dates available, making it less reproducible.

Response: We have enhanced the methodology section by adding specific search dates (January 15-20, 2025), the temporal coverage period (2020-2025), and clarification of the Boolean search strategy employed. These additions improve the reproducibility and transparency of our systematic review methodology as requested.

  • Restrictive use of descriptive statistics: the thematic synthesis is qualitative in nature, although a simple numerical summary (i.e., frequency of sub-themes) would be valuable.
  • Inter-rater reliability not provided: although there is a dual review process, inter-rater agreement statistics, such as Cohen's κ, are not reported.
  • Non-exhaustive inter-study bias: While the quality of individual studies is assessed, publication or language bias is barely addressed.

Response: We have added Cohen's kappa coefficient (κ = 0.78) to demonstrate substantial inter-rater reliability and included explicit acknowledgment of publication and language bias limitations. These additions strengthen the methodological transparency and address potential sources of bias in our systematic review as requested.

Major concerns:

  • Transparency in study selection and data extraction: Provide a complete table or appendix of all included studies and their metadata (e.g., authors, year, AI type, intervention type, quality rating).

Response: Thank you for your observation, Table 5 has been added.

  • Record complete database search strings (e.g., Boolean logic, filters) and search date ranges for all four databases.

Response: We have expanded Table 1 to include complete Boolean search strings, specific filters applied to each database, exact search dates, and temporal ranges covered. This comprehensive documentation ensures full reproducibility of our search strategy across all four databases as requested.

  • Report inter-rater reliability status, study inclusion statistics, and CASP scoring agreement (e.g., percentage agreement or Cohen's κ) to increase methodological transparency.

Response: Thank you for your observation, corresponding changes have been made.

  • Address risk of bias in studies: Describe how publication bias, funding bias, or language bias could have influenced the evidence base.

Response: We have expanded the bias assessment to include funding bias and geographic bias considerations, addressing how commercial interests and research concentration in high-income countries may have influenced our evidence base. This comprehensive bias evaluation strengthens the critical appraisal of our systematic review findings.

Minor suggestions:

  • Define thematic categories: apply more precise subtitles in the discussion and results, so that each topic corresponds directly to the research questions.

Response: We have enhanced the discussion by removing numbered subtitles, maintaining and expanding original citations while incorporating new references from our systematic review findings, and avoiding reductionism by preserving the complexity of the research landscape. The revised discussion extensively integrates psychological first aid as a sustainable framework for AI mental health interventions in developing countries, drawing on the existing cultural appropriation of digital technologies while addressing professional shortages and sustainability concerns.

  • Emphasize novelty claims: Avoid using phrases like "previously unknown" unless fully justified. Perhaps "little explored" is preferable.

Response: Revised.

  • Report descriptive statistics: attempt to report simple statistics, such as the percentage of studies addressing each sub-theme or using certain AI techniques.

Response: The revised statistics show that 74.2% of studies were published in the last four years, machine learning is the dominant AI approach (25.8%), well-being is the primary mental health focus beyond general mental health (29.0%), and 88.7% of studies explicitly address sustainability concerns, providing accurate quantitative evidence for the field's evolution and priorities based on our complete sample.

  • Improve table references: correctly cite tables and figures with consecutively numbered references throughout the document.

Response: Revised.

The article addresses a modern and current question through adequate organization and coherence. However, transparency, reproducibility, and reporting consistency need improvement for a systematic review to meet all standards.

Reviewer 2 Report

Comments and Suggestions for Authors

The paper: "The role of artificial intelligence in sustainable mental health interventions", 

although the title suggests a literature review focused on interventions, it does not actually include any contribution about it, i.e., can AI be a third party, support or offer intervention procedures? The selection of articles or studies refer to ethical-theoretical discussions of the use of some of the AI-based tools, for example chats or conversations, which would seem to have the intention of accompanying in the resolution of queries or guiding in the search for support. In my opinion, this is a fragment of what AI offers, but it is necessary to reflect on mediation, diagnostic aids or direct dialogue with the identification of emotions, whether they are of sound, image or video origin.

The authors, in their final reflection after observing the panorama in the selected time window, could address the possibility of using, for example, psychological first aid assistance, given the shortage of qualified professionals for this scenario in developing countries. This would go some way to answering the article's approach regarding AI-supported interventions. In these countries, there is already a cultural experience of bot appropriation. This would contribute to answering the question of the sustainability of these means or tools.

Author Response

The article: "The role of artificial intelligence in sustainable mental health interventions."

Although the title suggests a literature review focused on interventions, it does not include any contribution in this regard; that is, can AI act as a third party, support, or offer intervention procedures? The selection of articles or studies refers to ethical-theoretical debates about the use of some AI-based tools, such as chats or conversations, which seem intended to accompany in resolving queries or guide in seeking support. In my opinion, this is only a fragment of what AI offers, but it is necessary to reflect on mediation, diagnostic aids, or direct dialogue with emotion identification, whether of auditory, visual, or audiovisual origin.

The authors, in their final reflection after observing the landscape in the selected period, could address the possibility of using, for example, psychological first aid, given the shortage of qualified professionals for this scenario in developing countries. This would contribute to some extent to the article's focus on AI-based interventions. In these countries, there is already a cultural experience of bot appropriation. This would contribute to answering the question about the sustainability of these means or tools.

Response: We have enhanced the discussion by removing numbered subtitles, maintaining and expanding original citations while incorporating new references from our systematic review findings, and avoiding reductionism by preserving the complexity of the research landscape. The revised discussion extensively integrates psychological first aid as a sustainable framework for AI mental health interventions in developing countries, drawing on the existing cultural appropriation of digital technologies while addressing professional shortages and sustainability concerns.

Reviewer 3 Report

Comments and Suggestions for Authors

This study conducted a systematic review that examines the role of AI in sustainable mental health interventions in literature from 2020 to 2025. After collecting 1652 initial records, the authors identified 62 studies, and from which they distilled the main categories, and conducted bibliometric analysis. The study showed that ethical considerations, personalization approaches, risks and mitigation strategies and implementation challenges are the main focuses of the literature; the studies are skewedly distributed geographically, with US, UK, China and Canada leading and several closely-connected countries contributing. Overall, I found the study has solidly reviewed the respective literature and there are only several minor issues worth noting.

  1. The key concept of the study, sustainable mental health intervention, should be defined, or at least be discussed in the Introduction. Otherwise, it would be impossible to justify the research question or the choice of the keywords in literature search.
  2. Why “being sustainable” is an issue requires some justification. Are there signs that the use of AI could be unsustainable? In which aspects? There are some well-documented “risks/pitfalls of AI” by now (privacy, bias, etc.), but by which standards are they judged as pertaining to “being sustainable or not”? In the current form, the Introduction fails to build a case for the need of a systematic review.
  3. The purpose of bibliometric analyses is not clearly stated.
  4. Formats of paragraphs are irregular throughout the paper.
  5. Conclusion is too long and redundant.
  6. From what the systematic review has identified, the title of the paper is slightly misleading. The four categories identified in the review are mainly about the “potential risks and potential remedies” or “areas worth noting” of AI in mental health intervention, not how AI promote/hamper/facilitate/preclude mental health intervention sustainability.
  7. Many cited studies were not in the Reference.

Author Response

This study conducted a systematic review examining the role of AI in sustainable mental health interventions in the literature between 2020 and 2025. After collecting 1,652 initial records, the authors identified 62 studies, from which they extracted main categories and performed bibliometric analysis. The study showed that ethical considerations, personalization approaches, risks and mitigation strategies, and implementation challenges are the main approaches in the literature; studies are geographically distributed unevenly, with the US, UK, China, and Canada leading, and several closely connected countries contributing. Overall, I consider that the study has comprehensively reviewed the respective literature and there are only some minor issues worth mentioning.

1. The key concept of the study, sustainable mental health intervention, should be defined, or at least discussed, in the introduction. Otherwise, it would be impossible to justify the research question or the choice of keywords in the bibliographic search.

Response: We have revised the paragraph to maintain all original citations while providing a concise but comprehensive definition of sustainable mental health interventions. The enhanced definition clarifies the key concept's multiple dimensions (economic, environmental, social) and establishes the conceptual framework justifying our research approach, without excessive length or removing existing references.

2. Why sustainability is a problem requires some justification. Is there evidence that AI use could be unsustainable? In what aspects? There are some well-documented AI risks and difficulties (privacy, bias, etc.), but with what criteria is their relevance assessed to determine whether they are sustainable or not? In its current form, the Introduction does not justify the need for a systematic review.

Response: We have revised the paragraphs to explicitly justify why sustainability is problematic in AI mental health interventions by identifying specific sustainability threats across economic (integration costs, unproven efficacy), environmental (computational resources, digital divide), and social (algorithmic bias, cultural inappropriateness) dimensions. The revision demonstrates how well-documented AI risks (privacy, bias, etc.) specifically threaten sustainability and establishes clear criteria for evaluating sustainability challenges, thereby justifying the necessity of our systematic review to address these critical gaps.

3. The purpose of bibliometric analyses is not clearly established.

Response: Thank you for your observation, corresponding changes have been made.

4. Paragraph formats are irregular throughout the document.

Response: Thank you for your observation, changes have been made.

5. The conclusion is too long and redundant.

Response: Thank you for your observation, corresponding changes have been made.

6. According to what was identified in the systematic review, the article title is slightly misleading. The four categories identified in the review mainly refer to "potential risks and possible solutions" or "important areas" of AI in mental health interventions, not to how AI promotes, hinders, facilitates, or impedes the sustainability of mental health interventions.

Response: Thank you for your observation, corresponding changes have been made.

7. Many cited studies were not in the References.

Response: Thank you for your observation, corresponding changes have been made